# Application of Process Intensification in the Treatment of Pit Latrine Sludge from Informal Settlements in Blantyre City, Malawi

**DOI:** 10.3390/ijerph17093296

**Published:** 2020-05-09

**Authors:** Khumbo Kalulu, Bernard Thole, Theresa Mkandawire, Grant Kululanga

**Affiliations:** 1Department of Environmental Health, University of Malawi, Faculty of Applied Sciences, P/Bag 303, Chichiri, Blantyre 3, 312225, Malawi; 2Department of Physics and Biochemical Sciences, University of Malawi, Faculty of Applied Sciences, P/Bag 303, Chichiri, Blantyre 3, 312225, Malawi; bthole@poly.ac.mw; 3Department of Civil Engineering, University of Malawi, Faculty of Engineering, P/Bag 303, Chichiri, Blantyre 3, 312225, Malawi; tmkandawire@poly.ac.mw (T.M.); gkululanga@poly.ac.mw (G.K.)

**Keywords:** dewatering and sanitization, unplanted drying bed, sludge dewatering cycle time, sludge solids loading rate, sustainable sanitation

## Abstract

Many developing countries lack the infrastructure needed for the treatment of fecal sludge. One limitation in implementing available treatment options is the limited availability of land in the urban areas of these countries. This paper investigated the application of process intensification as a way of reducing the land area required to dewater and sanitize pit latrine sludge from informal settlements in Blantyre City, Malawi. The intensification of the sludge treatment process was achieved by enhancing dewatering through the application of additives and by combining the dewatering and sanitization stages. Nine combinations of sludge, lime and rice husk dosages, in addition to a control, were simultaneously loaded on unplanted drying bed units to dewater for 29 days. The study found a significant reduction of 21% to 73% in the land area required to dewater and sanitize pit latrine sludge. From the study, process intensification was shown to have the potential to significantly reduce the land area required to dewater and sanitize pit latrine sludge from informal settlements in Malawi cities. This makes it an option that can be implemented close to informal settlements, despite land limitation in these areas.

## 1. Introduction

Many developing countries lack appropriate fecal sludge treatment infrastructure [1,2]. The best method in most countries is to treat fecal sludge in existing centralized wastewater treatment systems, which, in most cases, have dysfunctional equipment and inadequate treatment capacity, leading to an appropriate-to-partial treatment of fecal sludge [3,4,5]. In addition, most of the centralized wastewater treatment plants are situated far from peri-urban or informal areas where fecal sludge is mostly generated. The foregoing leads to long haulage distances, which are associated with high transportation costs. The extended haulage time and cost to such centralized treatment facilities have been linked to the inability of much of the sludge to reach treatment facilities and end in urban environments in these developing countries [1]. The aforementioned situation still prevails in developing countries, despite the existence of technologies that are deemed to be suitable on the basis of their simplicity, reliability and robustness to shock loads, zero-to-low energy requirement, low skill requirement for operation and maintenance (O&M), very low O&M costs, reduced risks associated with system failure and increased reuse opportunities [6]. Such options include unplanted drying beds, planted drying beds, co-treatment in waste stabilization ponds, anaerobic pond settling–thickening tanks, constructed wetlands, lime stabilization, co-composting, extended storage and deep row entrenchment [2,7,8]. However, most of these require a large area of land to be implemented, which significantly contributes to the capital cost. Such a large area of land is rarely available close to the points of sludge generation. Process intensification provides an opportunity to reduce inputs into the pit latrine sludge treatment. Process intensification is defined as the attainment of a drastic increase in efficiency by using much less to produce much more [9]. Among the targeted outcomes of process intensification are the reduction in capital and running cost, land requirement, time, input materials, energy requirement, waste generation, nuisances, and risk and hazards [10,11]. These outcomes align well with the underlying concepts of sustainable sanitation, Most Appropriate Technology and sustainable sanitation [12,13,14,15]. According to Lutze et al. [16], process intensification might be attained through the “integration of unit operations, integration of functions, integration of phenomena and targeted enhancements of phenomena in a given operation”. Process intensification has been applied in biodiesel production to reduce long residence times, high operating costs and energy consumption, and low production efficiency [17]. Rao et al. [18] have demonstrated a reduction in the volume of rotating drying beds through process intensification. In membrane engineering, process intensification has been investigated in terms of energy reduction [19]. In wastewater treatment, process intensification has been used to come up with treatment units that are simple to construct and operate, compact, have relatively lower power consumption and lessen waste generation [20,21]. For example, studies such as Pekdemir et al. [20] and Keskinler et al. [21] have applied process intensification for wastewater ferrous iron removal and oxygen transfer, respectively. In fecal sludge treatment, a few studies such as Seck et al. [22] and Gold et al. [23] reflect some form of application of the principles of process intensification. However, most of these investigations have investigated process intensification at the level of targeted enhancement of phenomena within treatment stages in a sequential treatment process. Sitter et al. [24] suggested that such localized intensification leads to weak improvements in the whole process and they advocate for a holistic view when applying process intensification. This paper investigates the extent to which process intensification could lead to a reduction in the land area required to dewater and sanitize pit latrine sludge from informal settlements in Blantyre City, Malawi. Unlike most studies, this research investigation is a combination of the targeted enhancement of phenomena and functional integration forms of process intensification in the sludge treatment process. The study investigated a targeted dewatering of 40% total solids (TS) and sanitization of ≤1000 cfu/g TS for fecal coliforms and ≤3 cfu/4 g TS for *Salmonella,* as per the World Health Organization’s guidelines [25] In assessing land area reduction, this study also investigated the treatment time reduction and sludge solid loading rates that are attainable through the intensification of the pit latrine sludge treatment process.

## 2. Materials and Methods

### 2.1. Study Area

The pit latrine sludge investigated in this study was obtained from latrines in Ndirande Township, the most populous informal settlement in Blantyre City, Malawi [26]. It is also one of the townships where pit latrine emptying has been considerably promoted and adopted.

### 2.2. Experimental Design

The central composite design was used to design treatments of different combinations of lime (L) and rice husks (RH) dosages. This provided nine treatments and a control where no lime or rice husks were added. The selection of the lime dosage range of 10%–30% TS was guided by the range 10%–40% TS documented to be effective in pathogen reduction [27,28,29]. The dosage of rice husks (5%–15%) was selected in such a way that so that the proportion of rice husks would not exceed 50% of the total volume of sludge and additive mixture, so as to ensure that the treatment handles more sludge than additives. Table 1 presents the quantities of lime, rice husks and sludge in the treatments and the control.

The values in the table were established from an average of 31.5% TS and a sludge density of 1200 kg/m^3^ from Kalulu et al. [30] and Greya et al. [29]. A rice husk density of 112 kg/m^3^ was established in this study.

### 2.3. Drying Bed Prototype Design

The unplanted drying beds in the prototype used in this study were designed based on the typical field configuration recommended in the literature [8,31]. They were designed to accommodate 100 mm-thick layers of coarse gravel, medium gravel and sand and 200 mm sludge loading depths. Figure 1 shows the prototype, which is made of twelve drying beds.

The size of the loading surface for each drying bed on the prototype was 0.65 m × 0.65 m. This sizing was done in such a way as to ensure that all the treatments and the control could be loaded from the quantity in a single batch of sludge that is practically manageable to pump in a single day. From the interaction with manual pit emptiers, it was established that emptiers rarely exceed 1 m^3^ sludge from a single latrine in a day. Based on the 1 m^3^ sludge requirement, 15% of sludge volume loss (through removal of non-fecal matter such as plastics and fabrics) during loading and a sludge loading depth of 200 mm, the study came up with the final design of drying beds that had a loading surface of 650 mm × 650 mm. The 15% for non-fecal matter was chosen as the average of the range of 10%–20% established by Still [32]. Each of the drying beds on the prototype was designed to treat 84.5 L of treatment sludge at a time.

### 2.4. Loading and Sampling from the Intensified System

On the prototype, a fixed drying bed was assigned to each treatment and the control. Six cycles of dewatering were run for each of the treatments and the control. The number of cycles/replications was calculated using the one-way ANOVA sample size calculator in Minitab 17. The inputs for the calculations were 80% power, a sludge–solids loading rate standard deviation of 88.7 kg TS/m^2^/year and a maximum mean difference of 220 kg TS/m^2^/year. The maximum mean difference and the standard deviation were established from the solids loading rates reported in studies by Koné et al. [33] and Strande et al. [8]. Each dewatering cycle was 29 days long and consisted of 1 day for sludge loading and 28 days for sludge dewatering. The dewatering time of 4 weeks was chosen as the average of the time range, which was provided by Wang et al. [29], to attain 40% total solids. For each dewatering cycle, the treatments and control were assigned random numbers, ranging from 1 to 10. The assigned numbers guided the order of the loading of drying beds on the prototype. The drying bed assigned the number 1 was loaded first and the one assigned the number 10 was loaded last. Sludge samples were collected from the loaded beds for laboratory analyses on day 1 (day of loading), day 15 and day 29. The loading and sampling of the sludge from the drying bed prototype was done from November 2017 to December 2018. During this period, the atmospheric temperature ranged from 12 °C to 36 °C. Atmospheric humidity ranged from 11% to 100%. Wind speed ranged from 0 km/hr to 33 km/hr. During the rainy season, a shed with a light roof and open on the four sides was provided to prevent moisture from getting into the loaded beds and subsequent prolonged dewatering times.

### 2.5. Laboratory Analyses

Laboratory analyses were done to characterize drying bed filter materials, additives (lime and rice husks), raw sludge and sludge mixed with additives. Particle size distribution analysis for drying bed filter materials (sand, medium gravel, and coarse gravel) and additives (lime and rice husks) was done using the sieve analysis method in British Standards, BS 1377 [34]. The determination of rice husk density used the method in Water Research Commission, WRC 2137 [30]. Sludge samples from the treatments and the control were analyzed for pH, biochemical oxygen demand (BOD_5_), chemical oxygen demand (COD), total ammoniacal nitrogen (TAN), density, moisture content (MC), total solids (TS), total volatile solids (TVS), *E. coli* and *Salmonella*. For each loading cycle density, BOD_5_, COD and TVS were only analyzed on day 1 of the control sample to get the characteristics of the raw sludge. Analyses of pH, TAN, moisture content, total solids, density, *E. coli* and *Salmonella* were done for all the treatments and the control at the three sampling time points (day 1, day 15 and day 29) in the loading cycle. pH was analyzed using the potentiometric method [30]. Density was analyzed by the method in WRC 2137 [35]. BOD_5_ was analyzed using the titrimetric method in BS 6068: Part 1 [36]. COD was analyzed using the closed reflux titrimetric method [30]. TAN was analyzed using the titrimetric method [30]. Moisture content, total solids and total volatile solids were determined using gravimetric methods [37]. *E. coli* determination was done using the membrane filtration method [32]. *Salmonella* analysis was done using the plate count method in Association of Official Analytical Chemists, AOAC [37]. Volumetric concentrations were converted to wet weight concentrations by dividing them by the weight of sludge mixed with distilled water in the parameter analyses. Dry weight concentrations for the parameters were obtained by dividing the wet weight concentrations for samples by their corresponding total solids. All the analyses of the samples were performed in duplicate, and the average values are presented in this study.

### 2.6. Data Analysis

#### 2.6.1. Sieve Analysis/Particle Size Distribution

Analyses for particle size distribution for the drying bed filter materials and sludge additives were done using the particle size distribution curves obtained by the sieve analysis method. The percentages of the particles (by mass) less than a given particle size were plotted against the logarithm of the effective particle diameter. The distribution of the particle sizes was presented using a three-point specification consisting of D_10_, D_50_, and D_90_. This specification is considered to be complete and appropriate for most particulate materials [38]. D_10_ (effective size) is the maximum particle size of the smallest 10% of the particles. D_50_ is the sieve diameter through which 50% of the particles pass and 50% are retained. D_90_ describes the sieve diameter through which 90% of the particles pass and 10% are retained. The D values were obtained from the intercepts for 10%, 50% and 90% of the cumulative particle size distribution curves. The uniformity coefficients for the filter media and sludge additives were calculated by dividing D_60_ by the effective size (D_10_). D_60_ was obtained from the particle distribution curves of the materials.

#### 2.6.2. Determination of Sludge Dewatering Cycle Time

The determination of the sludge dewatering cycle time was based on the target of 40% total solids attainment in the sludge. This was chosen because it is the higher limit of solids content that is typically achievable through dewatering [2]. The determination of the time required to reach 40% TS in each treatment and the control was done using the logarithmic function presented in Equation (1). The logarithmic function was chosen based on the shape of the plots of the total solids against the time obtained in this study, as well as the findings from the studies by Smollen [39] and Seck et al. [22].
(1)$$\mathrm{TS}=k\ln\left(t_{d}\right)+c$$
where TS is total solids (%), t_d_ is the sludge dewatering time (days), and k and c are constants.

For each treatment and the control, the constants in the equation were calculated using Microsoft Excel Solver, using the total solids values for day 1, day 15 and day 29. With the constants determined, the Goal Seek function in Microsoft Excel was used to calculate the dewatering time for the treatments and the control to reach 40% TS. For each of the treatments and the control, the sludge dewatering cycle time was calculated using Equation (2).
(2)$$t_{dc}=t_{L}+t_{d}+t_{sr}$$
where *t_dc_* is sludge dewatering cycle time (days), *t_L_* is the sludge loading time (days)*, t_d_* is the sludge dewatering time (days) and *t_sr_* is the sludge removal time (days).

One-way ANOVA with Dunnett’s post hoc was used to compare the means of the treatments and the control for significant differences in the dewatering cycle time among the treatments and the control.

#### 2.6.3. Determination of Sludge Solids Loading Rates

Using the dewatering cycle times calculated from Equation (2), the solids loading rates for the treatments and the control were calculated using Equation (3).
(3)$$SLR_{s}=\frac{V_{s}*D_{s}*TS_{i}*365}{t_{dc}*A*100}$$
where *SLR_s_* is the sludge solids loading rate (kg/m^2^/yr), *V_s_* is the volume of the applied sludge (m^3^), *D_s_* is the density of raw sludge (kg/m^3^), *TS_i_* is the sludge initial solids content (%), *t_dc_* is the sludge dewatering cycle time to reach 40% TS (days), and *A* is the sludge loading surface area (m^2^).

One-way ANOVA with Dunnett’s post hoc was used to assess significant differences in the sludge solids loading rates in the treatments and the control.

#### 2.6.4. Treatment Land Area Requirement Reduction

The reduction in the land area requirement for each of the treatments, relative to the control, was calculated using Equation (4).
(4)$$Land\;area\;reduction\;\left(\%\right)=\left(1-\left(\frac{Control\;solids\;loading\;rate}{Treatment\;solids\;loading\;rate}\right)\right)*100$$

### 2.7. Ethical Approval

Ethical approval (no. P11/17/231) was obtained for the study from the National Commission of Science and Technology (NCST).

## 3. Results

### 3.1. Drying Bed Filter Material Characteristics

The effective size (D_10_) of the sand used in the drying bed was 0.25 mm. The D_50_ and D_90_ values for the sand were 0.53 mm and 1.6 mm, respectively. The D_60_ value for the sand was 0.6 mm. The uniformity coefficient (C_u_) for the sand was 2.4. The D_10_ of the medium gravel used in the middle layer of the drying bed filter unit was 5.2 mm. The D_50_ and D_90_ values for the middle layer gravel were 7 mm and 9.2 mm, respectively. The D_60_ value for the middle layer gravel was 7.5 mm. The C_u_ for the middle layer gravel was 1.44. The D_10_ of the coarse gravel used in the bottom layer of the drying bed was 8.5 mm. The D_50_ and D_90_ values for the bottom layer gravel were 15.5 mm and 19 mm, respectively. The D_60_ value for the bottom layer gravel was 16 mm. The value of the C_u_ for the bottom layer gravel was 1.88.

### 3.2. Sludge Additives Characteristics

The D_10_ of the rice husks was 0.9 mm. The D_50_ and D_90_ values for the rice husks were 1.7 mm and 2.4 mm, respectively. The D_60_ value for the rice husks was 1.8 mm. The C_u_ for the rice husks was 2. For lime, about 84% of the mass of the lime passed through the smallest sieve size of 0.038 mm used in this study. As such, it was not possible to determine the D_10_, D_50_ and D_60_ values for the lime. The D_90_ value for the lime was 0.05 mm.

### 3.3. Raw Sludge Characteristics

Table 2 presents the characteristics of the raw pit latrine sludge used in the process intensification study. BOD_5_ concentration ranged from 7452 mg/L to 15,653 mg/L, with a mean value of 11,107 mg/L. COD concentration ranged from 56,640 mg/L to 168,000 mg/L, with a mean value of 105,984 mg/L. The TAN concentration ranged from 139 mg/L to 201 mg/L, with a mean value of 170 mg/L. The dry weight concentration of TAN ranged from 0.9 mg/g TS to 3.4 mg/g TS, with a mean of 1.5 mg/g TS.

### 3.4. pH in Treatments and Control

On day 1, the mean pH in the control (8.3 ± 0.2) was lower than the means for the treatments, which ranged from 11.4 ± 0.1 to 13.8 ± 0.1. On day 15, the mean pH value in the control (7.9 ± 0.1) was also lower than the mean values for the treatments, which ranged from 10.4 ± 0.7 to 13.7 ± 0.1. On day 29, the mean pH value in the control (8.0 ± 0.2) was lower than the mean values for the treatments, which ranged from 10.8 ± 0.4 to 13.8 ± 0.1.

### 3.5. Total Ammoniacal Nitrogen

On day 1, the mean TAN in the treatments and control ranged from 0.42 ± 0.06 mg/g TS to 1.53 ± 0.4 mg/g TS. On day 15, the mean TAN in the treatments and control ranged from 0.32 ± 0.08 mg/g TS to 0.66 ± 0.17 mg/g TS. On day 29, the mean TAN in the treatments and control ranged from 0.22 ± 0.04 mg/g TS to 0.46 ± 0.12 mg/g TS.

### 3.6. Moisture Content

On day 1, the average moisture content in the control (85.3 ± 2.4%) was higher than the treatment, with the moisture content ranging from 62.9 ± 1.5% to 83.6 ± 1.4%. For day 15, the average moisture content in the control and treatments ranged from 51.9 ± 1.3% to 69.5 ± 2.1%. On day 29, the average moisture content in the control and treatments ranged from 44.3 ± 2.1% to 57.2 ± 2.2%.

### 3.7. Total Solids

On day 1, the average TS in the control (14.7 ± 2.4%) was lower than the treatment, with the average TS ranging from 16.4 ± 1.4% to 37.1 ± 1.5%. For day 15, the average TS in the control and treatments ranged from 30.5 ± 2.1% to 48.1 ± 1.3%. On day 29, the average TS in the control and treatments ranged from 42.8 ± 2.2% to 55.7 ± 2.1%.

### 3.8. E. coli

For all three sampling days, the study found *E. coli* in two treatments (L_10_, RH_5_ and L_6_, RH_10_) and the control. On day 1, the mean *E. coli* concentrations in the treatments L_10_, RH_5_ and L_6_, RH_10_ were 125,367± 62,092 cfu/g TS and 251,551 ± 105,845 cfu/g TS, respectively. The mean *E. coli* concentration in the control was 584,182 ± 208,465 cfu/g TS. On day 15, the *E. coli* concentrations in treatments L_10_, RH_5_ and L_6_, RH_10_ were 35,372 ± 15,831 cfu/g TS and 69,788 ± 26,568 cfu/g TS, respectively. The mean *E. coli* concentration in the control was 164,221 ± 56,266 cfu/g TS. On day 29, the *E. coli* concentrations in the treatments L_10_, RH_5_ and L_6_, RH_10_ were 23,712 ± 8074 cfu/g TS and 50,628 ± 14,663 cfu/g TS, respectively. The mean *E. coli* concentration in the control was 62,552 ± 11,968 cfu/g TS.

### 3.9. Salmonella

The study found *Salmonella* in three treatments (L_10_, RH_5_; L_10_, RH_15_ and L_6_, RH_10_) and the control at all three sampling times. On day 1, the mean *Salmonella* concentrations were 210 ± 210 cfu/g TS for treatment L_10_, RH_5_; 316 ± 316 cfu/g TS for treatment L_10_, RH_15_; 304 ± 304 cfu/g TS for treatment L_6_, RH_10_; and 1659 ± 518 cfu/g TS for the control. On day 15, the mean *Salmonella* concentrations were 85 ± 85 cfu/g TS for treatment L_10_, RH_5_; 80 ± 80 cfu/g TS for treatment L_10_, RH_15_; 152 ± 152 cfu/g TS for treatment L_6_, RH_10_; and 600 ± 77 cfu/g TS for the control. On day 29, the mean *Salmonella* concentrations were 80 ± 80 cfu/g TS for treatment L_10_, RH_5_; 36 ± 36 cfu/g TS for treatment L_10_, RH_15_; 41 ± 41 cfu/g TS for treatment L_6_, RH_10_; and 336 ± 33 cfu/g TS for the control.

### 3.10. Sludge Dewatering Cycle Time

Figure 2 shows the mean dewatering cycle times to attain 40% TS for the six replicates of the treatments and the control. The mean dewatering cycle times in the treatments and the control ranged from 5.5 days to 53 days.

A one-way ANOVA test showed a significant difference in the mean dewatering cycle times in the treatments and the control (F(9,50) = 3.2, *p* = 0.004). A Dunnett’s post hoc test revealed that treatments L_30_, RH_15_ (6.2 days), L_20_, RH_17_ (5.5 days) and L_10_, RH_15_ (6.2 days) had significantly shorter dewatering cycle times than the rest of the treatments and the control. The mean dewatering cycle times for the rest of the treatments ranged from 9.7 days to 44.5 days. The mean dewatering time for the control was 53 days. There was no statistically significant difference in the mean dewatering cycle times in the control and the other seven treatments.

### 3.11. Sludge Solids Loading Rates

Figure 3 shows mean sludge solids loading rates from the six replications of the treatments and the control. The mean sludge solids loading rates among the treatments and control ranged from 286 kg TS/m^2^/year to 1043 kg TS/m^2^/year.

A one-way ANOVA showed a significant difference in the mean sudge solids loading rates (F(9,50) = 3.7, *p* = 0.001) of the treatments and the control. A Dunnett’s post hoc test revealed that the sludge solids loading rates for treatments L_30_, RH_15_ (1043 kg TS/m^2^/year) and L_20_, RH_17_ (1013 kg TS/m^2^/year) were significantly higher than the rest of the treatments and the control. The mean sludge solids loading rates for the rest of the treatments and the control ranged from 286 kg TS/m^2^/year to 636 kg TS/m^2^/year. There was no statistically significant difference in the mean solids loading rates of the control and the other seven treatments.

### 3.12. Treatment Land Area Requirement Reduction

Figure 4 shows the mean percentage land area requirement reduction that the treatments would achieve in order to treat the same mass of solids in the same time as the control. The mean land area requirement reductions were established from six replications of the treatments and the control. The land area reductions resulting from the addition of lime and rice husks to the sludge ranged from 21% in treatment L_10_ and RH_5_ to 73% in treatments L_30_ and RH_15_.

## 4. Discussion

### 4.1. Physical and Hydraulic Characteristics of the Intensified System Materials

The bulk density of rice husks of 112 kg/m^3^ established in this study was comparable to the range of 90-110 kg/m^3^, which is presented in the literature [2]. The uniformity coefficients for the filter material, i.e., sand (2.4), middle layer medium gravel (1.44) and bottom layer coarse gravel (1.88) and rice husks (2), were within the recommended value of less than four for effective dewatering in a filter configuration. Values of the uniformity coefficient above four lead to abrupt changes in the hydraulic conductivity of the filter and promote subsurface pore clogging in the filter [40].

### 4.2. Total Time to Attain Dewatering and Sanitization

In this study, one rationale for the intensification of the sludge treatment process was that the intensification would lead to a reduction in the total time required to dewater and sanitize pit latrine sludge. The sanitization target in this study was to attain ≤1000 cfu/g TS for fecal coliforms and ≤3 cfu/4 g TS for *Salmonella* as recommended by the World Health Organization [25]. Except for three treatments (L_10_, RH_5_; L_10_, RH_15_ and L_6_, RH_10_) and the control, all the treatments had pathogen concentrations that met the World Health Organization’s recommendation by the time they reached 40% TS. Unlike the other treatments, these three treatments and the control had average pH values of less than 12 at all sampling times. Based on the treatments investigated in this study, a lime dosage of 20% TS was found to be the minimum required to attain sanitization. The 20% TS dosage is higher than the 10% TS that Abu-Orf et al. [28] found, and within the range of 10%-35% TS found by Greya et al. [29]. One possible explanation for these different results might be the differences in the levels of mixing done for the sludge and lime. Abu-Orf et al. [28] and Greya et al. [29], for example, used mixers to attain homogeneity, unlike manual mixing, which was employed in this study. It is highly probable that a higher quantity of lime was needed to make sure that it was uniformly distributed throughout the sludge–additive mixture in this study. In addition, although other studies have found ammonia to be key in pathogen die-off, the findings from this study tend to show that ammonia may not have been the major contributor to pathogen die-off as the total ammoniacal nitrogen concentrations established did not vary much among the treatments and the control at three sampling times within the cycles and throughout the sampling cycles. In some cases, pathogen die-off may, to a lesser extent, be attributed to a low moisture content, which fell below the minimum 50% that Bakare et al. [4] stated to be adequate for microbial activity.

The findings of the study demonstrate that the intensified system has the potential to reduce on time in order to attain both dewatering and sanitation. In this study, the total treatment time to attain these two objectives was found to be shorter than the time that it took the control to attain 40% total solids dewatering alone. The dewatering time of 53 days established for the unconditioned sludge in the control in this study was higher than the dewatering times reported on unconditioned sludge in the literature. Wang et al. [41], for example, reported dewatering times in the range of 14 days to 42 days to attain 40%–45% total solids. Seck et al. [22] found dewatering times ranging from 8 days to 12 days. The shorter dewatering times in these studies may be due to a higher level of stabilization in the sludge they investigated, which may have led to a higher dewaterability than the sludge in this study. Another possible explanation for the difference may be the smaller loading depths and lower sludge solids loading rates (in the range of 100 kg TS/m^2^/year to 150 kg TS/m^2^/year) that were investigated by Seck et al. [22]. However, a study by Kuffour et al. [42] found similarly shorter dewatering times, which ranged from four days to seven days for unconditioned sludge, despite having a similar sludge loading depth of 200 mm, as was the case in this study. The shorter drying time might have resulted from a lower total solid target during dewatering. In the study by Kuffour et al. [42], the dewatering cycle ran up to the time when the sludge had become spadable and stopped producing percolate, which in most cases is within the range of 20%–25% TS.

While the treatment times found in this study are comparatively similar to the dewatering times in the control and other studies where the process was not intensified, it should be kept in mind that the treatment time in this study covers both dewatering and sanitization. The difference between the treatments and the control in this study, therefore, provides a picture of the minimum time that could be saved from the treatments that were investigated in this study. Factoring in the additional time required for the sanitization of dewatered sludge from the control in this study would likely lead to a longer total treatment time. The additional time would range from days to years, depending on the sanitization option chosen. Compared to the maximum of 44.5 days established in this study, it would take more than six months to attain the targeted dewatering and sanitization through extended storage [43,44]. Composting would require a minimum of 58 days to attain the targeted dewatering and sanitization levels, based on a time requirement of 5 days to 15 days, for composting, as reported by Tayler [2].

### 4.3. Sludge Solids Loading Rates

The design of the study theorized that shortening the dewatering cycle time through process intensification would lead to an increased sludge loading frequency and a resulting increase in the sludge quantity treated in a unit bed area in a given time. The findings of this study demonstrate that higher sludge solids loading rates were attainable through the intensification of the fecal sludge treatment process. All treatments investigated in this study had higher sludge solids loading rates than the control. The sludge solids loading rate (286 kg TS/m^2^/year), established for the control in this study, falls within the range of 50 kg TS/m^2^/year to 420 kg TS/m^2^/year, which was reported for unconditioned sludge in the literature [8,45]. The sludge solids loading rates range for the treatments in this study (361 kg TS/m^2^/year to 1043 kg TS/m^2^/year) was lower than the range of 505 kg TS/m^2^/year to 1239 kg TS/m^2^/year, which was established in a study by Kuffour [46]. In his study, he conditioned sludge with different dosages of sawdust and loaded to a similar depth of 200 mm on drying bed columns. One possible explanation for the lower range of sludge solids loading rates in this study could be the longer dewatering cycle times required to attain a higher dewatering target of 40% total solids, in comparison to the 20%–25% TS dewatering target by Kuffour [44]. Dewatering cycle time is used when calculating the sludge solids loading rates and holding loading depth, sludge density, total solids concentration and loading area constant—the shorter the dewatering cycle time, the higher the sludge solids loading rates, as more loading cycles are achievable in a given time.

### 4.4. Treatment Land Area Requirement Reduction

The major underlying hypothesis for the process intensification study was that it was possible to reduce the total land area that is required to treat pit latrine sludge from informal settlements in Malawi cities. This study demonstrates that process intensification has the potential to lead to significant reductions in the land area required to dewater and sanitize pit latrine sludge. The treatment land area reduction range (21% to 73%) was slightly lower than the range of 59%–97% reduction in the drying bed area established in a study by Gold et al. [23]. This difference may be due to the lower dewaterability of the pit latrine sludge used in this study, in comparison to the more stabilized septage conditioned by Gold et al. [23]. The differences in the dewaterability may also arise from the difference in the levels of mixing and homogeneity of the conditioner–sludge mixtures. Considering the small volumes that were thoroughly mixed in jars by Gold et al. [20], it is highly likely that the sludge–conditioner mixture was more homogeneous in comparison to this study in which about 84 L of sludge and conditioners were mixed manually. In the study by Gold et al. [23], the larger reduction in land area requirement could have resulted from the shorter time required to attain the dewatering target for composting, which Koné et al. [32] presented as 20% TS. It is highly probable that targeting 40% TS dewatering in their study may have resulted in longer dewatering times and corresponding smaller land area reductions. As with the dewatering cycle time, the range of land area requirement reduction attained in the different treatments, relative to the control, represents the minimum reductions to achieve dewatering and sanitization. Reductions of larger magnitudes are likely to be attainable if the land area required to sanitize the dewatered unconditioned sludge was considered.

The reduction in land area requirement for dewatering and sanitization achieved in this study demonstrates that the intensified system provides a sludge treatment option that has potential to be implemented closer to the informal settlements where pit latrine sludge is generated and land availability is limited. The shorter distances achieved by having such systems closer to points of sludge generation could significantly reduce the haulage distance between the point of generation and treatment. According to Taweesan et al. [1] shorter distances may imply a reduction in haulage time and cost, with an increased likelihood of getting more sludge reaching treatment facilities. Since transportation costs constitute the major cost in the fecal sludge management chain, the reduction of this cost may translate to a significant reduction in the cost of the different services in the chain and could make the services affordable to more onsite sanitation users in informal settlements. This has the potential to decrease the improper disposal of pit latrine sludge into urban environments and an associated reduction in the public and environmental health risks associated with the sludge being in urban areas of developing countries. A reduced distance has the potential to lead to lower fuel usage and lower vehicle maintenance costs. In addition, this also has the potential to reduce the carbon footprint that arises from the haulage of sludge for treatment, disposal and reuse. In addition, the use of a substitute material, where rice husks are not readily available, provides an opportunity to address the challenge of some of the biodegradable solid wastes in Malawi cities.

## 5. Conclusions

This study demonstrates that the application of process intensification (lime and rice husk dosing) in the treatment of pit latrine sludge has the potential to lead to a significant reduction in the time or land area required to dewater and sanitize pit latrine sludge from informal settlements in Malawi cities. This makes the intensified system an option that can be implemented close to the informal settlements with limited available land. There is a need to investigate the intensification of the sludge treatment process in terms of all three treatment objectives of dewatering, sanitization and stabilization. Materials such as locally made activated carbon could be included as one of the additives, or a layer in the filter system to enhance the stabilization of organics. Future studies on the intensification of the pit latrine sludge treatment process should consider the lime dosage range of 20%–30% TS and a rice husks dosage range of 15%–17% TS as the best guess and to pick other treatments around these levels.

## Figures and Tables

**Figure 1 ijerph-17-03296-f001:**
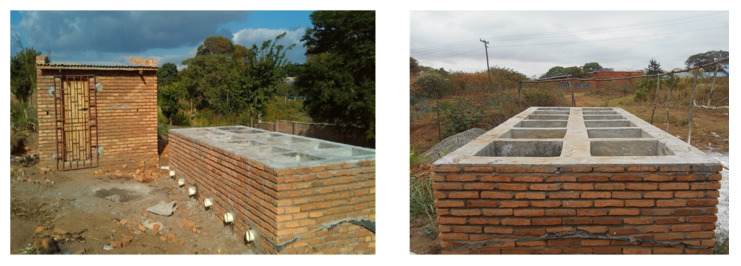
Unplanted drying beds prototype.

**Figure 2 ijerph-17-03296-f002:**
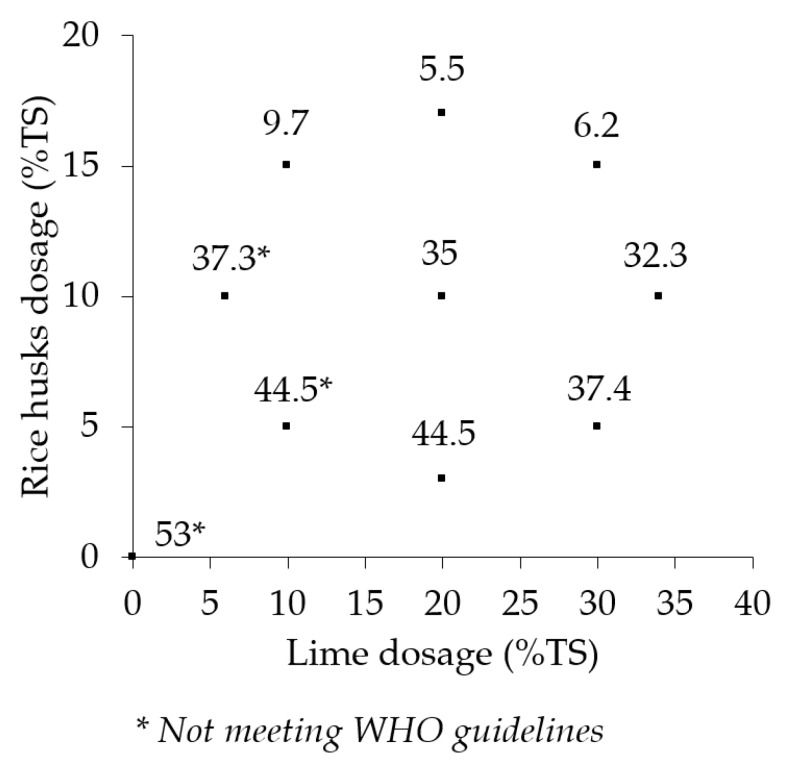
Sludge dewatering cycle time (in days) for the treatments and the control. WHO: World Health Organization.

**Figure 3 ijerph-17-03296-f003:**
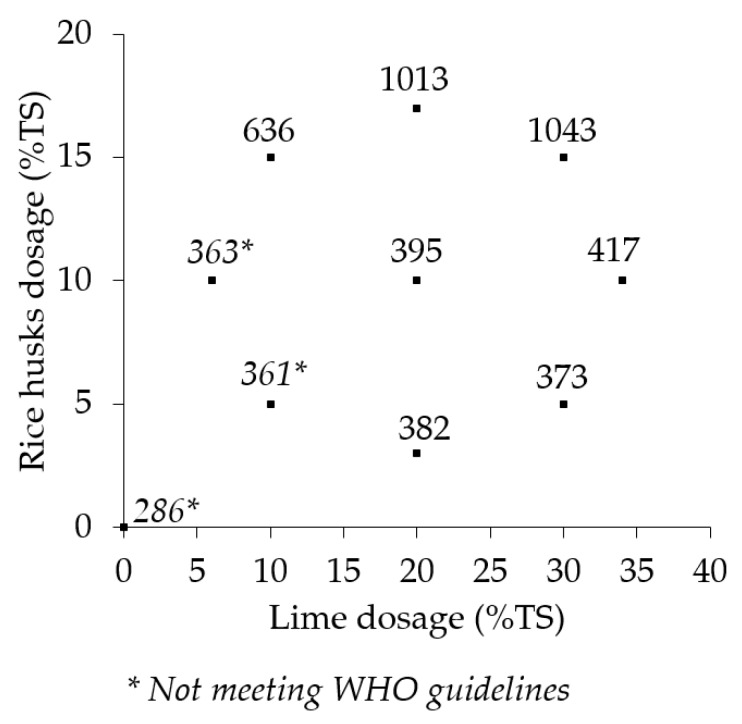
Sludge solids loading rates (in kg TS/m^2^/year) in the treatments and control.

**Figure 4 ijerph-17-03296-f004:**
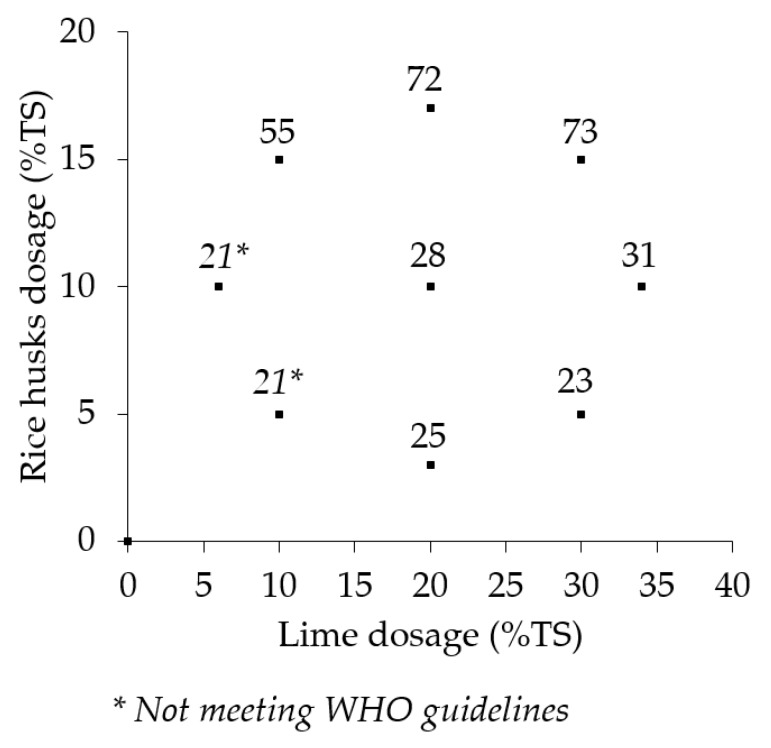
Percentage land area requirement reduction relative to the control.

**Table 1 ijerph-17-03296-t001:** Details of the treatments and control. L: lime; RH: rice husks.

Treatment	Lime Dosage (% TS *)	Rice Husks Dosage (% /TS)	Lime Quantity (kg)	Rice Husks Quantity (kg)	Rice Husks Volume (L)	Volume of Sludge (L)
L_10_, RH_5_	10	5	3.15	1.6	14	70
L_10_, RH_15_	10	15	3.15	4.7	42	42
L_30_, RH_15_	30	15	9.46	4.7	42	42
L_30_, RH_5_	30	5	9.46	1.6	14	70
L_6_, RH_10_	6	10	1.89	3.2	28	56
L_34_, RH_10_	34	10	10.72	3.2	28	56
L_20_, RH_3_	20	3	6.31	0.9	8	76
L_20_, RH_17_	20	17	6.31	5.4	48	37
L_20_, RH_10_	20	10	6.31	3.2	28	56
Control	0	0	0	0	0	85

* TS: Total solids.

**Table 2 ijerph-17-03296-t002:** Summary statistics of raw sludge characteristics. BOD: biochemical oxygen demand; COD: chemical oxygen demand; TAN: total ammoniacal nitrogen; MC: moisture content; TS: total solids; TVS: total volatile solids.

Parameter	Mean	Min	Max
pH	8.3	8.0	9.0
Density (kg/m^3^)	1107	1085	1115
BOD_5_ (mg/l)	11,107	7452	15,653
COD (mg/l)	105,984	56,640	168,000
TAN (mg/g TS)	1.5	0.9	3.4
MC (%)	85.3	77.5	94.7
TS (%)	14.7	5.3	22.5
TVS (mg/g TS)	506	452	540
*E. coli* (cfu/g TS)	584,182	160,780	1,504,514
*Salmonella* (cfu/g TS)	1659	80	3134

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
