# Peer review of "Application of Process Intensification in the Treatment of Pit Latrine Sludge from Informal Settlements in Blantyre City, Malawi"

_ijerph, 2020, doi:10.3390/ijerph17093296_

Round 1

Reviewer 1 Report

Comments are included in the commented pdf file attached 

Reviewer 2 Report

This paper investigated the extent which process intensification could lead to a reduction in land area required to dewater and sanitise pit latrine sludge from informal settlements of Blantyre City in Malawi. In assessing the land area reduction, the study also investigated the treatment time reduction and solids loading rates attainable through the intensification of the pit latrine sludge treatment process. It needs minor revision before it is published in this journal. The following issues should be carefully addressed.

  1. There are some grammatical errors and the sentences are monotonous in the manuscript, the authors need to go through the entire manuscript sentence by sentence;
  2. Abstract should be improved since it can not adequately attract the readers in this area;
  3. Authors should clearly mention the novelty of the study in Introduction;
  4. The figure number needs to be revised.
  5. Please add error bars and indicate how many repetitions in figures.
  6. Adding some discussion on the process intensification used in other fields.

Round 2

Reviewer 1 Report

The presentation has improved greatly.

However, I have one more comment: as the Authors have carried out the experiments in open environment, they should specify which were the  environmental conditions during the experiments, such as, at least, temperature and rain, if any.

The Authors may add a comment about the effets of intense and prolonged rain. Should they think that some kind of light roof or cover (open on the 4 sides, like a gazebo or canopy) on the drying beds should be advisable?
